# ^1^H-NMR Karplus Analysis of Molecular Conformations of Glycerol under Different Solvent Conditions: A Consistent Rotational Isomerism in the Backbone Governed by Glycerol/Water Interactions

**DOI:** 10.3390/ijms24032766

**Published:** 2023-02-01

**Authors:** Yoshihiro Nishida, Reina Aono, Hirofumi Dohi, Wuxiao Ding, Hirotaka Uzawa

**Affiliations:** 1Molecular Chirality Research Center, Department of Applied Biological Chemistry, Institute of Environmental Horticulture, Chiba University, Matsudo 271-8510, Japan; 2Nanomaterials Research Institute, National Institute of Advanced Industrial Science and Technology (AIST), 1-1-1 Higashi, Tsukuba 305-8565, Japan

**Keywords:** glycerol, rotational isomerism, conformation, Karplus equation, ^1^H NMR spectroscopy, bilateral symmetry, cell membrane, glycerolipids

## Abstract

Glycerol is a symmetrical, small biomolecule with high flexibility in molecular conformations. Using a ^1^H-NMR spectroscopic Karplus analysis in our way, we analyzed a rotational isomerism in the glycero backbone which generates three kinds of staggered conformers, namely gt (*gauche*-*trans*), gg (*gauche*-*gauche*), and tg (*trans*-*gauche*), at each of *sn*-1,2 and *sn*-2,3 positions. The Karplus analysis has disclosed that the three rotamers are consistently equilibrated in water keeping the relation of ‘gt:gg:tg = 50:30:20 (%)’ at a wide range of concentrations (5 mM~540 mM). The observed relation means that glycerol in water favors those symmetric conformers placing 1,2,3-triol groups in a *gauche/gauche* geometry. We have found also that the rotational isomerism is remarkably changed when the solvent is replaced with DMSO-*d_6_* or dimethylformamide (DMF-*d_7_*). In these solvents, glycerol gives a relation of ‘gt:gg:tg = 40:30:30 (%)’, which means that a remarkable shift occurs in the equilibrium between gt and tg conformers. By this shift, glycerol turns to also take non-symmetric conformers orienting one of the two vicinal diols in an antiperiplanar geometry.

## 1. Introduction

Glycerol (**1**) is a small, symmetric biomolecule with high molecular flexibility in conformations (Figure 1) [1,2,3]. Free rotation along each of *sn*-1,2 and *sn*-2,3 C-C bonds in the backbone generates three different staggered rotamers with vicinal diols in either *gauche*-*trans* (gt), *gauche*-*gauche* (gg), or *trans*-*gauche* (tg) geometry (Figure 1). In combinations of these three rotamers, the backbone rotational isomerism totally gives a set of nine different conformations [1,2], which are traditionally denoted with Greek letters (α, β, and γ) as shown in Figure 1. When a pair of enantiomeric conformers like αβ and βα can be dealt identical to each other, the nine conformers are converged to six different conformers (αα, αβ, αγ, γγ, γβ, and ββ) [4,5].

The rotational isomerism in the glycerol backbone has been extensively examined in both experimental and simulation studies [1,2,3,4,5,6,7,8,9,10,11,12,13,14,15], which include a characterization study on Li^+^ glycerates used in a lithium battery [6]. In history, a glycerol crystal was first obtained by Koningsveld [2] and reported to adopt a symmetric αα-conformer with gt/gt rotamers. The symmetric molecular conformation is confirmed in a more recent high-resolution X-ray diffraction analysis by Kusukawa et al. [7]. The rotational isomerism occurs in both the liquid and solution phases [4,5], while the proportions of the nine or six different conformers are inconsistent between neutron diffraction and simulation studies [8,9,10]. For glycerol in the dilute aqueous state close to biological conditions, 220 MHz ^1^H-NMR spectroscopic analysis was conducted at a wide range of temperatures (−5~95 °C) in D_2_O solution by Koningsveld [3]. The NMR analysis has shown that glycerol in water exhibits a consistent rotational isomerism in the backbone independent of the temperatures examined. The backbone rotational isomerism proposed in the authentic ^1^H-NMR study had strong impact on the later stereochemical study of glycerol in the solution state [13,14,15].

With interests in the structure and biological functions of cell membrane glycerolipids [16,17,18], we have once a conformational behavior of glycerol (**1**) in water (D_2_O) with 400 MHz ^1^H-NMR spectroscopy in the characterization study of chirally deuterated *sn*-glycerols [19,20]. The studies were extended to the ^1^H-NMR Karplus analyses of sorts of glycerol esters coupled with circular dichroic spectroscopy [16,21] and applied in enzymatic reactions [22,23,24]. In our ^1^H-NMR Karplus analysis, all the glycerol C-H protons are unambiguously assigned by using chirally mono-deuterated *sn*-glycerols. We have confirmed thereby that glycerol in water (D_2_O) and symmetric glycerol tri-esters in organic solvents like CDCl_3_ and CDOD_3_ exhibit a common rotational isomerism in the backbone, which keeps a relation of ‘gt > gg > tg (%)’ in the dilute solution phase [16,17,18].

In our ^1^H-NMR Karplus analysis of glycerol (**1**), the three kinds of the staggered rotamers (%) are calculated with the scalar ^1^H-spin coupling constants (^3^*J* Hz) of glycerol methylene protons (H*R* and H*S*, Figure 1) applied in a general Karplus equation proposed by Haasnoot et al. [25]. On the other hand, our previously reported ^1^H-NMR data of glycerol (entry 1, Table 1) is notably deviated from the authentic ^1^H-NMR study (entry 3). Including the other spectral data stored in HMDB database (entry 2) [26], the ^1^H-NMR spectral data of glycerol make some deviations at ^1^H-coupling constants (entries 1~3, Table 1). In an extensive simulation and 500 MHz ^1^H-NMR spectroscopic study by Callam et al. [13], the authentic ^1^H-NMR spectral data is ascertained. Therefore, glycerol in water is believed to exhibit the rotational isomerism as reported in the authentic 220 MHz ^1^H-NMR spectroscopic study even in the later simulation studies.
3.1 gt + 2.8 gg + 10.7 tg = ^3^*J* _H2,H1S_ (Hz)(1)
10.7 gt + 0.9 gg + 5.0 tg = ^3^*J*
_H2,H1R_ (Hz)(2)
gt + gg + tg = 1(3)

We doubt that the deviation of the ^1^H-coupling constants between entries 1 and 2 (Table 1) may arise from differences in ^1^H-NMR experimental conditions such as the concentration of glycerol and/or DSS (sodium salt of trimethylsilyl propanesulfonate) [27]. In the authentic 220 MHz ^1^H-NMR study, the concentration of glycerol was set at 5 wt.% in D_2_O (540 mM, entry 3, Table 1), while the other two ^1^H-NMR data (entries 1 and 2) are measured at ca. 100 mM. Though DSS is commonly used as an internal standard, no particular attention has been paid to the effect of this ionic reagent on the rotational isomerism of glycerol.

In the present study, we analyze glycerol in D_2_O solutions without using DSS for the purpose of removing ambiguities in our former ^1^H-NMR Karplus study of glycerol. Concentrations are changed between 5 mM and 540 mM in D_2_O to examine effects of substrate concentrations. The solvent is also changed from D_2_O into organic solvents like DMSO-*d_6_* and DMF-*d_7_* to examine the possible effect of solvents. The present study clearly shows that glycerol in water (D_2_O) solutions adopts a distinctive, consistent conformation property that is very close to that of our former study (entry 1) rather than that of the authentic one (entry 3). The study also indicates that the backbone rotational isomerism is marginally changed in the presence of DSS and the other chaotropic reagent (guanidine hydrogen chloride), while it is dramatically changed in organic solvents (DMSO and DMF).

## 2. Materials and Methods

### 2.1. Glycerol in D_2_O Solutions for ^1^H-NMR Karplus Analysis

Anhydrous glycerol was dissolved in 99.8% D_2_O to make a 5 wt.% glycerol solution in D_2_O (540 mM). This solution was kept at room temperature for several hours and then diluted with D_2_O into 5 mM, 54 mM, and 108 mM solutions. In another experiment, glycerol was mixed with 10 molar equivalents of H_2_O, kept for several hours (140 min) at room temperature, and diluted with D_2_O to give a D_2_O solution of aqueous glycerol (500 mM). Preparations of glycerol solutions (50 mM) containing an equimolar amount of guanidine hydrogen chloride, aqueous glycerols in 99.5% DMSO-*d_6_* or 99.5% DMF-*d_7_* are conducted in the same way as those mentioned above.

### 2.2. ^1^H-NMR Spectroscopic Measurement and Spectral Data Acquisition

A ^1^H-NMR spectrum of each of the glycerol solutions was measured in a glass tube (5 mm Φ) on JEOL 500 MHz or Bruker 400 MHz machine at 19–20 °C. For the analysis of glycerol in D_2_O solutions, a 500 MHz ^1^H-NMR instrument (JEOL JNM-ECA 500, Tokyo, Japan) was operated under a good shim, and the obtained FID signals Fourier transformed into ^1^H-NMR spectral data with a Delta 2 program (JEOL). Fundamental ^1^H-NMR spectral data (chemical shifts and coupling constants) were assembled in the first order analysis and checked with a second order ABX analysis and multi-spin ^1^H-simulation with an algorithm of Castillo et al. [28]. The first order analysis was performed on the JEOL computer program expanding each of ^1^H-signals to determine a weight-averaged center. In this way, the chemical shifts (δ = ±0.0005 ppm) and ^1^H-coupling constants (*J* = ±0.05 Hz) were obtained with high accuracy. In those analyses without using internal standards like TMS and DSS, chemical shifts are determined with the signal of HOD (δ_HOD_ = 4.7620 ppm) or CHCl_3_ (δ_CHCl3_ = 7.2600 ppm) as referential solvent signals.

### 2.3. ^1^H-NMR Karplus Analysis of Glycerol in the Dilute Aqueous Phase

A general ^1^H-NMR Karplus equation of Haasnoot et al. [25] is extended to each of Equation (1) (eq 1) and Equation (2) (eq 2) according to our reported manner [16,29]. The electronegativity factors (*Ri*) are set at 1.3 (O1 and O2) and 0.4 (C2) in both eq 1 and eq 2. The two equations have different dihedral angles (Φ) for vicinal diols. In eq 1, the three staggered rotamers (gt, gg, and tg) are assumed to have ideal angles (Φ = ±60 and 180 degrees) with respect to 1,2-diols. In eq 2, the dihedral angles of gt and gg rotamers are set at ±65 degrees [16,29]. Applying the ^3^*J_1,2_* (Hz) values of each H1*R* and H1*S* proton in these equations, the time-averaged populations (%) of gt, gg, and tg are obtained.

**(a)** **A general Karplus equation [25]:**^3^*J_H1,H2_* (Hz) = 13.22cos^2^Φ − 0.99cosΦ + ∑ R*i* (O1, C2, O2),
where R*i* = Δχi {0.87 − 2.4cos^2^(ζΦ + 19.9∣Δχi∣)}

**(b)** 
**Equation (1) (+60, −60, 180 in degrees) = eq 1 [16]:**


3.1gt + 2.8gg + 10.7tg = ^3^*J* _H1S,H2_ (Hz)

10.7gt + 0.9gg + 5.0tg = ^3^*J* _H1R,H2_ (Hz)

gt + gg + tg = 1

**(c)** 
**Equation (2) (+65, −65, 180 in degrees) = eq 2 [16]:**


2.5gt + 2.3gg + 10.7tg = ^3^*J* _H1S,H2_ (Hz)

10.2gt + 1.3gg + 5.0tg = ^3^*J* _H1R,H2_ (Hz)

gt + gg + tg = 1

#### System of Equations

With high resolution ^1^H-NMR machines (Figure 2), glycerol dissolved in D_2_O gives a pair of double-doublet signals at δ 3.55 ppm and δ 3.65 ppm (Table 1), which are assigned to H1*R* (or H3*S*) and H1*S* (or H3*R*), respectively [19,20].

Figure 2b show that (*1S*)-[1-^2^H]-*sn*-glycerol gives no signal from the glycerol H1*S* proton in ^1^H-NMR spectroscopy, while the remaining H1*R* proton gives a broad doublet (*bd*) signal. Since glycerol makes the relation between H1*R* and H3*S* as well as between H1*S* and H3*R* magnetically identical each other, all the glycerol C-H protons are assigned as shown in Table 1 and Figure 2. Matteson et al. [30] reported the same assignment as ours for glycerol in a D_2_O solution in the absence of DSS, meaning that there is no ambiguity in the ^1^H-spectroscopic assignment of all the glycerol C-H protons in D_2_O solutions regardless the presence or the absence of DSS.

## 3. Results and Discussion

### 3.1. ^1^H-NMR Spectroscopic Profile of Glycerol in D_2_O Solutions without DSS

^1^H-NMR spectra of glycerol in D_2_O solutions usually give a strong ^1^H-signal of HOD as one of solvent species. The HOD signal, which comes from the deuterium (^2^H) exchange of the deuterated solvent (D_2_O) with OH groups in both glycerol and contaminated H_2_O, is used in the present study as standard (δ = 4.7620 ppm) to determine chemical shifts. The ^1^H-signals of glycerol C-H protons shift upfield (3.40~3.80 ppm) compared to the HOD signal (Figure 3a).

Here, it is noteworthy that the HOD signal is gradually broadened with increasing concentrations of glycerol in D_2_O as shown with an inserted graph in Figure 3a. This phenomenon is accelerated when the concentration exceeds 100 mM. This observation is rationalized by the physicochemical property of glycerol which can enhance the viscosity of water solutions making hydrogen (H) bonding interactions with the surrounding water [31,32,33,34]. The enhanced viscosity shortens the spin—spin relaxation time (T2) of the HOD signal and makes the ^1^H-signal broader [35,36]. It is also possible that the increased viscosity may slow the ^2^H-exchange to make the D_2_O solution of glycerol heterogeneous. In a theoretical study of glycerol-water interactions, Laaksonen and coworkers [8] have stated in conclusion that the presence of water increases the overall mobility of glycerol while glycerol slows the mobility of water. This statement also accords with our experimental result (Figure 3) that the ^1^H-signals of glycerol C-H protons are not so largely broadened compared to the ^1^H-signal of HOD.

To seek an underlying mechanism involved in the above observation, pristine glycerol was treated with 10 molar amounts of H_2_O molecules for several hours, diluted with D_2_O into a 500 mM solution, and then analyzed with ^1^H-NMR spectroscopy to give a very unique ^1^H-NMR spectrum as shown in Figure 4. The aqueous glycerol treated with H_2_O in advance turns to give a sharp ^1^H-signal to the solvent species (HOD) with a half-width comparable with the spectrum of glycerol at 100 mM in D_2_O. Taking a careful look at a base line, the aqueous glycerol also gives an extraordinarily broad ^1^H-signal between 5.3 and 5.8 ppm region in D_2_O. Judging from the ^1^H-NMR chemical shifts of pristine glycerol reported by Wexler et al. [37], the broad ^1^H-signal may be assigned to those hydroxy protons (ROH) being involved in interactions between glycerol and water. Probably, the initially added H_2_O are involved in the highly broad signal together with the ^1^H-signals of glycerol OH protons.

As described above, the ^1^H-NMR spectroscopy of glycerol exhibits intriguing phenomena in D_2_O solutions without DSS, indicating that glycerol changes the physicochemical property of the surrounding water. This property supports our idea that the ^1^H-NMR spectroscopic data of glycerol may be changed by concentrations, DSS, and other extrinsic factors.

### 3.2. Effect of Concentrations on the Conformational Behavior of Glycerol in Water (D_2_O)

The 540 mM solution was diluted with D_2_O (>99%) without DSS into each of 108 mM, 54 mM, and 5 mM solutions, and each of the glycerol solutions was applied in our ^1^H-NMR Karplus analysis with 500 MHz ^1^H-NMR spectroscopy (entries 1~4, Table 2). The results of entries 1~4 (Table 2) indicate that the ^1^H-signals of glycerol C-H protons shift upfield with increasing concentrations in D_2_O. A difference in chemical shifts (Δ ppm) shows that each of the glycerol C-H protons makes a parallel upfield shift each other with increasing concentrations. On the other hand, none of scalar ^1^H-spin coupling constants (^2^*J* and ^3^*J* Hz) and relative chemical shifts make change. This means that the rotational isomerism determined as the time-averaged populations (%) of gt, gg, and tg rotamers is kept unchanged at the range of concentrations examined (5 mM~540 mM in D_2_O).

In our ^1^H-NMR Karplus analysis, the three rotamers (%) are calculated with each of Equation (1) (eq 1) and Equation (2) (eq 2) (Section 2). These two equations are obtained from the general Karplus equation [25]. In eq 1, ideal dihedral angles (+60, −60, 180 degrees) are assumed in each of the three staggered rotamers, and this equation is widely applied in the stereochemical study of carbohydrates [24,29,38,39,40] and glycerols [20,21,41,42]. In eq 2, the dihedral angles are tuned by ±5° for each of two *gauche* conformers (gt and gg) assuming possible deviations from the ideal dihedral angles [16,29]. We have seen such a case that eq 1 gives minus populations (<0%) to tg when 1,2-diols takes an extraordinary rotation mode in which only gt and gg conformers are allowed in equilibrium. The minus populations are adjusted by tuning the dihedral angles (Φ) in the general Karplus equations or by using eq 2 or other equations as described in our former studies [16,29].

The populations of the three rotamers are calculated with each eq 1 and eq 2, and the results in entries 1~5 indicate that glycerol in D_2_O favors gt more strongly than gg and tg in the backbone rotational isomerism. The backbone rotation gives the three rotamers an empirical relation of ‘gt > gg > tg = ca. 50:30:20 (%)’. This relation is rigidly maintained in D_2_O regardless of the concentrations examined in the present study (5 mM~540 mM). All the three ^1^H-NMR spectral data of glycerol measured in the presence of DSS (entries 1~3, Table 1) also follow the empirical relation, though they make substantial deviations in the ^3^*J* (Hz) values of each other. Judging from this result, the internal standard (DSS) makes no significant change in the rotational isomerism of glycerol in D_2_O solutions as far as used in a limited amount.

From the preceding observations that the HOD and ROH signals are so broadened (Figure 3 and Figure 4), we predicted that glycerol might show a conformational change in the backbone with increasing concentrations. However, the results of entries 1~5 show that glycerol shows no apparent change in the backbone conformation even at the highest concentration (540 mM). In entry 5, glycerol was treated in advance with 10 molar excess amounts of water (H_2_O) before the ^1^H-NMR spectroscopic measurement in D_2_O (Figure 4). In this experiment, we tried to examine a possible influence of bound water (H_2_O) not of bound D_2_O on the conformational behavior of glycerol. Though the ^1^H-signal of the solvent (HOD) signal was sharpened by this experiment to the level of glycerol at 100 mM, no difference was induced in the backbone rotation isomerism. This result indicates that D_2_O and H_2_O as solvents cause no difference from each other in the conformational behavior of glycerol.

According to simulation studies for aqueous glycerols by Egorov et al. [8] and Chen and Li [11], the backbone rotational isomerism is hardly changed by glycerol concentrations in water. Though the proposed rotational isomerism is notably different from each and those from ours, the conformation property of glycerol is strongly controlled in interactions with surrounding water. The bound water may make a shelter to interfere with glycerol/glycerol interactions even at high concentrations over 500 mM.

### 3.3. The Backbone Rotational Isomerism of Glycerol in the Dilute Aqueous Phase Generating the Nine or Six Different Conformers

Glycerol in water gives the rotational isomerism among the nine or six different conformers as mentioned before (Figure 1). The rotational isomerism can be approximated with the multiplication products of the three staggered rotamers (gt, gg, and tg (%)) [3,42]. In the current ^1^H-NMR Karplus analysis, the three rotamers (%) are obtained with each of Equation (1) (eq 1) and Equation (2) (eq 2) as described above. The overall rotational isomerism is calculated from each of the two equations, and the results are denoted with eq 1 and eq 2 in Table 3.

The results by eq 1 and eq 2 are compared to result eq-1 (A) and result eq-2 (A) which are obtained with the 220 MHz ^1^H-NMR data given in the authentic study (entry 5, Table 2). A rotational isomerism actually reported in the authentic 220 MHz ^1^H-NMR study is given as result A [3,14]. Out of simulation studies of aqueous glycerol, results by Callam et al. [13] (result B), Yongye et al. [14], (result C), and Jeong et al. [15] (result D) are also listed together. Our result by eq 1 deviates from result A by 4~5% at most of conformers except for ββ and γγ. These deviations arise not only from the difference in the ^3^*J* (Hz) values of H*R* and H*S* signals but also from differences in the limit ^3^*J* values (Hz) employed in the calculation of the three rotamers (%). This is because results eq-1 (A) and eq-2 (A) still remain similar deviations. Result eq-2 (A) is, however, very close to result A. This means that Equation (2) (eq 2) has the limit ^3^*J* (Hz) values very close to those applied in empirical equations used in the authentic 200 MHz ^1^H-NMR spectroscopic study.

In simulation studies in literature, glycerol in both liquid [4,5] and aqueous phases (>40 wt.%) [8] favor the symmetric αα-conformer as the most preferred conformation, followed by each of αγ/γα conformers in an enantiomeric relation. In the dilute aqueous phase, the two simulation studies (B and C) commonly show that the symmetric αα-conformer is decreased to 20% or less. These simulation studies may be largely influenced by the authentic NMR study (result A). Of the three simulation studies, the latest one (result D) by Jeong et al. [15] is closest to our results using the Karplus-based Equation (1) (eq 1) and Equation (2) (eq 2). In the MD and ^1^H-NMR simulation study, Woods and coworkers [14] also produced a rotational isomerism very close to ours. They had proposed result C that is close to the authentic study (result A) rather than to ours.

The overall rotational isomerism can be better understood with a three-dimension (3D) distribution map. In Figure 5, our results are expressed with 3D maps in which the nine different conformers are distributed in a symmetrical fashion with respect to glycerol *sn*-1,2 and *sn*-2,3 positions. These maps give a diagonal line, on which three symmetric conformers of αα (gt/gt), γγ (gg/gg), and ββ (tg/tg) are distributed. The relation of ‘gt > gg > tg = 50:30:20 (%)’ is maintained at each of horizontal (*sn*-1,2) and vertical (*sn*-2,3) axes reflecting the bilateral symmetry of glycerol.

The 3D distribution map in Figure 5a shows that glycerol favors the four kinds of conformers in the order as ‘αα (gt/gt) > αγ/γα (gt/gg) > gg (gg/gg)’ in equilibrium. All these conformers comprise of *gauche/gauche* rotamers (gt and gg) orienting the glycerol 1,2,3-triols in a symmetrical *cis*-relation. On the other hand, the 3D distribution map in Figure 5b shows glycerol can also take the enantiomeric pair of αβ (gt/tg) and βα (tg/gt) conformers. Including the pairs of gβ (gg/tg) and βg (tg/gg) conformers, these conformers have an antiperiplanar vicinal diols at either the *sn*-1,2 or *sn*-2,3 position, which makes glycerol non-symmetric in a sense of molecular conformations. The probability of these non-symmetric conformers amounts to 35% in calculation with eq 2 (23% + 12%, Table 3). In the results B and D in Table 3 as well as a neutron diffraction study by Towery et al. [9,10], glycerol is predicted to adopt these non-symmetric conformations beside the symmetric ones in equilibrium. In the nine different conformers, a ββ (tg/tg) conformer is the least populated. This result is commonly shown in all the three simulations studies (results B, C, and D, Table 3) and rationalized in former studies in terms of 1,3-*syn*-diaxial or 1,3-syperiplanr interaction. An analogous interaction is reported in monosaccharides, particularly along the C5–C6 bond in hexopyranoses (Figure 6) [29,43]. Glycerol in the ββ (tg/tg) conformation gives a stereochemical circumstance close to the exocyclic position of D-glucopyranose (D-Glc*p*) with a tg rotamer (Figure 6a). The D-Glc*p* C5–C6 bond strongly disfavors a tg rotamer due to the 1,3-*syn*-diaxial interaction to show an extraordinary rotational isomerism between gt and gg rotamers. The D-Glc*p* type of di- and oligosaccharides take over this property in water solutions [38,39,40,44].

A similar correlation is available with D-Gal type of mono- and oligosaccharides with an axial C4-OH bond (Figure 6b). Their exocyclic C5–C6 bonds predominantly adopt a gt rotamer in water, while the rotational isomerism is randomized in DMSO causing a notable equilibrium shift from gt to tg [29,43]. This shift is observed for *O*-acylated D-galactopyranoses in CDCl_3_, DMSO-*d_6_*, acetone-*d_6_*, and pyridine-*d_5_*. Different from the D-Glc type of saccharides, they retain the gg rotamer in equilibrium at 15~20% regardless of the repulsive 1,3-synperiplanar interaction, kinds of the solvents, and the notable equilibrium shift between gt and tg caused by solvents. There is a stereochemical factor which keeps the proportion (%) of the gg conformer nearly constant.

### 3.4. Conformational Behaviors of Glycerol under Different Solvent Conditions

As mentioned above, the relation of ‘gt > gg > tg = 50:30:20 (%)’ in equilibrium represents a distinctive conformation property of glycerol in water at concentrations below 540 mM. Judging from our previously reported ^1^H-NMR spectral data of glycerol measured in D_2_O with DSS [20] (entry 1, Table 1), the conformation property is not affected by DSS as far as applied in a small amount, for examine, less than 0.05 mM in D_2_O in our experiments. However, we still suspect the conformational behavior can be changed by certain extrinsic factors such as ionic salts and solvents. In the current study, we have examined effects by guanidine hydrochloride which is known as a chaotropic reagent [45] and also those by DMSO and DMF which are widely used as aprotic organic solvents (Table 4). To examine these effects, glycerol was initially dissolved in D_2_O at 500 mM, and the solution kept at room temperature (20~22 °C) for several hours and then diluted with solvents to each of 50 mM solutions for ^1^H-NMR Karplus analysis.

In Table 4, entry-2 gives the ^1^H-NMR spectral data of glycerol in D_2_O (50 mM) containing guanidine hydrogen chloride (1:1 molar ratio). The guanidine reagents are known to act as chaotropic reagents on proteins and nucleic acids [45]. It is also reported that a cytoplasm membrane protein called aqua(glycero)porin is reported to utilize an Arg guanidino group for transportation of glycerol across membranes [46,47,48]. The ^1^H-NMR data of entry 2 shows that this reagent causes a notable shielding shift for all the glycerol C-H protons when the HOD signal is used as a standard of chemical shifts. This change is, however, not accompanied by shift in the rotational isomerism. This result suggests that the upfield shifts may be due to the downfield shift of the HOD signal. Similar to glycerol/glycerol interactions in water, glycerol/guanidine interactions may be effectively blocked by water. This solvent can make solid interactions with each of glycerol 1,2,3-triol and guanidino amino groups to allow glycerol to keep the relation of ‘gt:gg:tg = 50:30:20 (%)’ even in the presence of the chaotropic reagent.

On the other hand, the results in entries 3 and 4 indicate that DMSO-*d_6_* and DMF-*d_7_* used as solvents notably induce a significant change. They cause a significant change not only in the chemical shifts of glycerol C-H protons but also in the backbone rotational isomerism. It is worth noting that a notable shift arises in equilibrium between gt and tg rotamers toward tg while leaving the proportion of gg almost unchanged. Two 3D distribution maps in Figure 7 show that the backbone rotational isomerism is totally randomized in these solvents. The empirical relation of ‘gt > gg > tg’ among glycerols is no longer maintained when analyzed with Equation (2) (Figure 7b).

The notable shift in equilibrium between gt and tg conformers is analogous to what we have seen for the D-Gal*p* type of sugars as mentioned above (Figure 6b). Ethylene glycerol (ethane-1,2-diol) and propane-1,2-diol are reported to exhibit analogous changes in conformations by solvents [49,50,51]. The behavior of alkane-1,2-diols favoring a *gauche* conformation in water, CHCl_3_, and pyridine solutions is explained in different ways like those with intramolecular OH/OH interactions and cooperative H-bonds [49,50,51] other than ‘*gauche effects*’ [52,53,54]. Conformations are changeable by interactions with solvents applied; water and alcohols with OH groups tend to stabilize a *gauche* relation for 1,2-diols, and this trend is applicable to the D-Gal type of hexopyranoses. DMSO and DMF with carbonyl (-C=O) groups tend to destabilize the *gauche* relation of 1,2-diols similar to the D-Gal types of sugars. Though glycerol is thought to largely follow the rotational isomerism of alkane-1,2-diols and the D-Gal type of saccharides, an explicit mechanism waits for advanced theoretical and experimental studies. The use of nuclear Overhauser effects (NOE or NOESY) and other advanced NMR techniques coupled with the current ^1^H-NMR Karplus analysis may provide a clear insight. 

## 4. Conclusions

We have seen in the present ^1^H-NMR Karplus analysis that glycerol in the dilute aqueous state exhibits the consistent backbone rotational isomerism which can be represented by a relation of ‘gt > gg > tg = ca. 50:30:20 (%)’. This relation is hardly changed in a wide range of concentrations in water (5 mM~540 mM) and even in the presence of DSS or the chaotropic reagent, guanidine hydrogen chloride. When the overall rotational isomerism of the six kinds of conformers is approximated with this relation, the result has disclosed a notable difference from those reported in the authentic ^1^H-NMR study and followed by the later simulation studies. We assume that the authentic study might have failed to obtain the ^1^H-NMR spectral data of glycerol H*R* and H*S* signals in correct ways. With a low resolution ^1^H-NMR spectrometer, the ^1^H-signals of glycerol methylene protons (H*R* and H*S*) often collapse in a close region to make the ^1^H-NMR Karplus analysis difficult to perform with high accuracy. We often experience this problem even in recent studies using 500 MHz ^1^H-NMR spectrometer. Molecular conformations of glycerols and other biomolecules analyzed with low resolution ^1^H-NMR spectrometry should be cautiously approached, particularly when applied as experimental evidence in simulation studies.

We have also seen in the present study that glycerol in water tightly attracts surrounding water molecules, reconstructs H-bond networks, makes the aqueous solution viscous, and establishes the consistent conformational behavior in water solutions. The rotational isomerism is remarkably changed in DMSO or DMF solutions. This change is indicative of the underlying mechanism that glycerol/water interactions largely contribute to the conformational behavior of glycerol in water solutions. On the other hand, we have no explicit explanation for the rotational isomerism of glycerol which shows the relation of ‘gt > gg > tg = ca. 50:30:20 (%)’ consistently in water and changes the relation into ‘gt:gg:tg = ca. 40:30:30 (%)’ in the two organic solvents. We hope theoretical approaches will give a clear answer using our precise ^1^H-NMR spectroscopic data and the Karplus calculations offered in the present study.

Glycerol is a kind of prochiral compound with respect to *sn*-1 and *sn*-3 position [55]. The prochirality has a great meaning in both the metabolism and catabolism of glycerol and glycerolipids in which enzymes like glycerol kinases and dehydrogenases differentiate between the glycerol *sn*-1,2 and *sn*-2,3 positions [56,57,58]. Unfortunately, none of these enzymatic reactions are fully understood in correlation with the molecular conformation of glycerol despite a close study is reported in the study of aqua(glycero)porin as cited in the text. Glycerol and glycerolipids have again begun to attract strong interest from the aspect of emergency of life [59,60,61] beside their potential as biofuels. Biochemistry and biomolecular science have many things to do with this important class of biomolecules well into the future.

## Data Availability

Additional information may be available in email contacts with Y.N. (Chiba Univ.) or H.U. (AIST) following rules by each of MDPI, Chiba Univ., and AIST.

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
