# Peer review of "1H-NMR Karplus Analysis of Molecular Conformations of Glycerol under Different Solvent Conditions: A Consistent Rotational Isomerism in the Backbone Governed by Glycerol/Water Interactions"

_ijms, 2023, doi:10.3390/ijms24032766_

Round 1

Reviewer 1 Report

Dear Authors,

I have reviewed your manuscript, and I am expressing my positive feedback. Your study is important for the readers of the International Journal of Molecular Sciences, and the obtained results are promising. In my opinion, there are a few corrections that would improve the overall quality of your manuscript:

·        Occasionally, there is no space between the numerical value and the unit. Please correct this throughout the whole text.

·        You keep writing “lit. number” throughout the text, for example, “lit. 19”. Instead, you should write “reference 19” or “references [10-15]”.

·        Within explanations for Table 1, the dashed lines between the equation and equation number should be removed.

·        Some schematics of Figure 1 seem to be too small. Is it possible to enlarge Figure 1?

·        Equations presented in Scheme 2 are not Scheme; they are equations. Maybe it would be better to name “Scheme 2” as a “System of equations”?

·        The conclusion is the weakest part of this manuscript. It is too large and difficult to follow. Currently, it contains a lot of results and mentions of references, which should not happen in this chapter. In this chapter, you should have emphasized the most important findings of your study, the implications of your study, and the potential directions to continue within your further studies. Please rewrite it, and transfer some parts of the conclusion to the results and discussion section.

Once you address all of the above-mentioned comments, I will gladly review your manuscript again.

Best regards

Author Response

Response is given in an attached file.

Reviewer 2 Report

It is a well written manuscript concerning to the conformer of glycerol in different solvent. The experiment, analyses and comparison with other work were performed carefully. 

Two small question:

1. It would be nice to comment a bit more seriously the simulation data (B,C,D) for water? Are the force filed appropriate?

2. Can you connected this results with the competition of intra and intermoleculare H-bond capability og Gly? (DMSO,DMF only H-bond acceptor, water: H-bond donor)

Author Response

Response is given in an attached file.

Reviewer 3 Report

In this manuscript titled by "1H-NMR Karplus analysis of molecular conformations of glycerol under different solvent conditions: A consistent rotational isomerism in the backbone governed by glycerol/water interactions ' - the by methods of  NMR spectroscopic Karplus approach to analyze a rotational isomerism in the glycerol backbone is used. The contents of the reviewed manuscript are very well described by the title. The authors showed that different solvents can shift conformational equilibrium.
In my point, this manuscript is well-written and could be recommended for publication in IJMS after a corresponding revision in which the following issues are addressed:
The experiments are not very well explained and the discussion is not clear enough. The finding reason for conformational change is an important issue.  Please add more description and discussion interaction of conformational effect for different solvents.
-The authors have to shed light on the similarities and differences among their work and the literatures of the problem.  A clear explanation, what is the new result in their work, and how it is build up upon previous work in the field. For example why you cant use the NOESY experiment for your analysis? Please see recnt reserch.
Interproton distance determinations by NOE - Surprising accuracy and precision in a rigid organic molecule (2011) Organic and Biomolecular Chemistry, 9 (1), pp. 177-184. DOI: 10.1039/c0ob00479k
The Role of Hidden Conformers in Determination of Conformational Preferences of Mefenamic Acid by NOESY Spectroscopy
(2022) Pharmaceutics, 14 (11),  â„– 2276, . DOI: 10.3390/pharmaceutics14112276
Conformational preferences of tolfenamic acid in DMSO-CO2 solvent system by 2D NOESY (2022) Journal of Molecular Liquids, 367, â„– 120481, . DOI: 10.1016/j.molliq.2022.120481

Author Response

Response is given in an attached file.
